# Patient, hospital and country-level risk factors of all-cause mortality among patients with chronic heart failure: Prospective international cohort study

**Benedetta Pongiglione**[1]*, **Aleksandra Torbica**[1,2], **Chris P. Gale**[3,4,5], **Luigi Tavazzi**[6], **Panos Vardas**[7,8], **Aldo P. Maggioni**[6,9]

1 Centre for Research on Health and Social Care Management (CERGAS), SDA Bocconi, Milan, Italy, 2 Department of Social and Political Sciences, Bocconi University, Milan, Italy, 3 Leeds Institute for Cardiovascular and Metabolic Medicine, University of Leeds, Leeds, United Kingdom, 4 Leeds Institute for Data Analytics, University of Leeds, Leeds, United Kingdom, 5 Department of Cardiology, Leeds General Infirmary, Leeds, United Kingdom, 6 Maria Cecilia Hospital—GVM Care & Research, Cotignola, Italy, 7 University General Hospital of Heraklion Second Surgical Department Heraklion, Heraklion, Greece, 8 Hygeia, Mitera, HHG Hospitals Group, Athens, Greece, 9 EURObservational Research Programme, European Society of Cardiology, Sophia-Antipolis, France

* benedetta.pongiglione@unibocconi.it

**Data Availability Statement:** This study involves third-party data that the authors do not have the rights to share. Data cannot be shared publicly

## Abstract

### Background

Although many studies have described patient-level risk factors for outcomes in heart failure (HF), health care structural determinants remain largely unexplored. This research reports patient-, hospital- and country-level characteristics associated with 1-year all-cause mortality among patients with chronic HF, and investigates geographic and hospital variation in mortality.

### Methods and findings

We included 9,277 patients with chronic HF enrolled between May 2011 and November 2017 in the prospective cohort study European Society of Cardiology Heart Failure Long Term registry across 142 hospitals, located in 22 countries. Mean age of the selected outpatients was 65 years (sd 13.2) and 28% were female. The all-cause 1-year mortality rate per 100 person-years was 7.1 (95% confidence interval (CI) 6.6–7.7), and varied between countries (median 6.8, IQR 5.6–11.2) and hospitals (median 7.8, IQR 5.2–12.4). Mortality was associated with age (incidence rate ratio 1.03, 95% CI 1.02–1.04), diabetes mellitus (1.37, 1.15–1.63), peripheral artery disease (1.56, 1.27–1.92), New York Heart Association class III/IV (1.91, 1.60–2.30), treatment with angiotensin-converting enzyme inhibitor and angiotensin receptor antagonists (0.71, 0.57–0.87) and HF clinic (0.64, 0.46–0.89). No other hospital-level characteristics, and no country-level healthcare characteristics were associated with 1-year mortality, with case-mix standardised variance between countries being very low (1.83e-06) and higher for hospitals (0.372).

because of confidentiality. Data are available from the European Society of Cardiology for researchers who meet the criteria for access to confidential data. To apply to get access to the dataset used for this analysis, the Heart Failure Long-Term Registry Patient Characteristics, please contact the European Society of Cardiology (Route des Colles, Les Templiers — CS 80179 Biot, 06093 Sophia Antipolis Cedex, France; email: eorp@escardio.org) indicating the relevant variables as reported in the case report form attached as supplementary file S3 in S1 Appendix.

**Funding:** The author(s) received no specific funding for this work.

**Competing interests:** The authors have declared that no competing interests exist. All authors have completed the ICMJE uniform disclosure form and declare: no support from any organisation for the submitted work; no financial relationships with any organisations that might have an interest in the submitted work in the previous three years; no other relationships or activities that could appear to have influenced the submitted work.

## Conclusions

All-cause mortality at 1 year among outpatients with chronic HF varies between countries and hospitals, and is associated with patient characteristics and the availability of hospital HF clinics. After full adjustment for clinical, hospital and country variables, between-country variance was negligible while between-hospital variance was evident.

## Introduction

Heart failure (HF) is characterized by a high rate of hospital admissions and death, significant functional compromise, reduced quality of life, and increased caregiver burden [1,2]. Remarkable progress in the treatment of HF has been made in the last few decades and included in the current International guidelines [3,4], with an improvement in survival of patients with chronic HF [5,6]. Several evidence-based trials have identified effective medical treatments for patients with HF and reduced ejection fraction; such treatments are currently recommended by current clinical guidelines and variably incorporated in clinical practice [5,6]. A study using data from the European Society of Cardiology's (ESC) Heart Failure Long-Term Registry (HF-LT-Registry, version 2013) found heterogeneity of treatments, most ineffective on hard endpoints, for patients with acute HF, while drug treatments for patients with chronic HF can be considered adherent to recommendations of current guidelines, even if dosing often appears too parsimonious [7].

Research has highlighted the considerable differences in HF outcomes between different countries [8,9]. Risk factors for HF outcomes have been studied mostly considering patient's clinical and socio-demographic characteristics. Age, medical history, comorbidities such as pulmonary, liver, and kidney disease, are generally known to be related with a higher risk of readmission [10] and mortality [11]. Other studies found socioeconomic factors, such as low health literacy [12] and poor social support [13], are associated with higher all-cause mortality among patients with HF.

Yet, hospital-level and country-level factors for HF outcomes remain largely unexplored. One of the few studies that considered hospital characteristics as a predictor of hospital re-admission found that discharge from hospitals with HF services is associated with lower readmission at both 7 days and 30 days [10]. Recent work [14] studied income inequalities within countries and HF outcomes, and found that greater inequality was associated with worse HF outcomes. The structure and organization of healthcare systems and hospitals may play an important role in the application of guideline recommendations in HF management and, as a consequence, in determining differences in patients' outcomes [15]. There is a growing interest in studying the association between country-level inequality, such as income, and various population health measures, but only a few studies have considered cardiovascular diseases.

This work aimed to fill this gap. Combining an international prospective cohort study, the ESC Heart Failure Long Term (HF-LT) Registry, version 2016, and an international ESC Atlas of cardiology, we created a unique set of data that enabled us to consider patient, hospital and country characteristics at once and explore their association with the all-cause mortality of patients with chronic HF. More specifically, we aimed to i) investigate between-country and hospital variation in mortality rates among patients with chronic HF; ii) identify the characteristics of patients, hospitals and countries associated with 1-year mortality of patients with chronic HF.

## Methods

### Design and setting

We combined information independently collected by two ESC projects, the prospective cohort study called ESC HF-LT-Registry and the ESC Atlas of Cardiology, and created an enhanced dataset to assess the concurrent association of patient', hospital' and country's characteristics with mortality in patients with chronic HF. Combining these databases, we were able to gather information at three levels hierarchically ordered: patient, hospital and country.

The ESC HF-LT-Registry was launched in May 2011 to describe the clinical epidemiology of outpatients and inpatients with HF and the diagnostic/therapeutic processes applied to these patients across European and Mediterranean countries. The ESC HF-LT Registry also includes a 'Site Questionnaire' where data on admission hospitals are collected, so providing information on both patients and hospitals. A detailed description of the dataset (version 2016) is available elsewhere [11].

The ESC Atlas of Cardiology is a collection of cardiovascular data across 57 countries managed by ESC to understand the structural determinants of cardiovascular disease outcomes. It utilizes multiple data sources, including for example the World Health Organization and the World Bank, to document risk factors, prevalence, and mortality of cardiovascular disease as well as national economic indicators. It also includes an ESC-sponsored survey data of health infrastructure and cardiovascular service provision provided by the national societies of the ESC member countries [16,17]. Two Atlas editions have been issued and include indicators collected between 2013 and 2017 [16,18]. We used both issues so to include a broader set of indicators, available in the years in which patient and hospital data were collected, spanning from 2013 to 2017.

This study complies with the Declaration of Helsinki. Participation in the ESC-HF-LT-R had been approved by each local institutional review board in accordance with its country's legislation. National Coordinators were responsible for obtaining the approval of the local review boards for this registry, if necessary. The Scientific Secretariat and Data Management team distributed the relevant documents in English (protocol, case report form, consent form) to the National Coordinators, who were responsible thereafter for their translation and adaptation to local standards. The full names of the Institutions of each National Coordinator in which the protocol was approved is reported in the supplementary file S1.1. All participants provided written informed consent (see file S2). No data were collected before the patient received detailed information and gave signed informed consent. No ethical approval was needed for this specific retrospective analysis that was carried out under a signed agreement between the Centre for Research on Health and Social Care (CERGAS) Bocconi University and the European Society of Cardiology.

### Study population

Outpatients aged over 18 years with chronic HF had been included in the HF LT Registry since May 2011. We considered for the analysis all participants who underwent the 12-months follow-up visit, which was performed to collect information on morbidity and mortality.

### Clinical characteristics

At baseline visit, extensive information on patients' characteristics were collected. We selected those variables known to be related with HF outcomes [11]. These included demographic characteristics (age and sex); information on cardiovascular risk factors collected at time of enrolment (body mass index (BMI) treated as a five-category variable (<20, 20–24, 25–29, 30–

34, >35), systolic blood pressure); clinical history (HF history, documentation of ischemic heart disease, diabetes, atrial fibrillation, peripheral artery disease (PAD); chronic obstructive pulmonary disease (COPD), chronic kidney dysfunction, left ventricular ejection fraction); physical data included the New York Heart Association (NYHA) class, pulmonary rales, hepatomegaly, peripheral oedema and third heart sound (S3 Gallop); information on medications comprised the use of angiotensin-converting enzyme (ACE) inhibitors, angiotensin receptor blockers (ARBs), beta blockers prior and/or during outpatient visit and device therapy.

## Outcome

The vital status of patients was recorded in the follow-up visit, which was expected to take place approximately one year after the baseline visit. Owing to between-country differences in the starting date of enrolment, there were varying follow-up times in the entire study group. We considered all deaths recorded in the follow-up visit, and censored survival time at day 365 for those patients whose follow-up visit took place more than a year after their admission.

## Hospital and country characteristics

The information on hospital's characteristics that we extracted from the ESC-HF-LT Site Questionnaire consisted in indicators of hospital's services for HF: whether there are catheterization and electrophysiological laboratories, cardiology services in site available 24 hours, units specifically dedicated to patients with HF and whether the hospital performs heart transplants.

The country characteristics selected from the ESC Atlas of Cardiology included indicators of socioeconomic status of a country: GDP per capita at purchasing price parity and the inequality Gini index, which ranges from 0 (perfect income equality), to 100 (perfect inequality); indicators of health care resources including health expenditure per capita and as percentage of GDP; life expectancy at birth and number of deaths for cardiovascular diseases per million inhabitants. We also classified countries based on their health care system and distinguished i) Beveridge-type (national health system) including Cyprus, Denmark, Greece, Italy, Portugal and Spain; ii) Bismarck-type (social health insurance) including Austria, Bosnia & Herzegovina, Bulgaria, Egypt, France, Israel, Lithuania, Serbia and Turkey, iii) Systems in transition (former Semaschko model) including Belarus, Czech Republic, Estonia, Hungary, Poland, Slovakia and Slovenia [19].

## Statistical analysis

Baseline characteristics were described using numbers and percentages for categorical data and means and standard deviations for normally and nonnormally distributed continuous variables.

To measure the association between 1-year all-cause mortality and selected risk factors, we conducted a hierarchical analysis and selected piecewise exponential survival model (PWE) and discrete time survival model [20], incorporating cluster-specific random effects (random intercept) to account for within-cluster homogeneity in outcomes. In PWE model, the time scale is divided into intervals and the hazard function is assumed to be constant within each interval [21,22]. We split the survival time into four trimesters given that the hazard of death among outpatients with chronic HF is stable over time and homogeneous within such time segments [11]. From the PWE model we obtained Poisson regression coefficients, whose exponential gave the incidence rate ratios that express the rate ratio change in mortality.

We considered the PWE model as the main model and run a discrete time survival model as supplementary analysis (more details in the S1 Appendix). From the multilevel logistic

model we assessed the degree of variation in mortality rates attributed to the country and hospital levels (and not explained by the model), using the intraclass correlation coefficient (ICC), as the calculation of the ICC from a multilevel Poisson model is not trivial [23,24].

We implemented several PWE model specifications in a stepwise fashion, considering the clinical model with only patient's characteristics as baseline and then investigating the additive value of hospital and country characteristics adding each level-specific characteristics separately (i.e., only patient's characteristics, patient's and hospital's characteristics, patient's and country's characteristics and full model). All statistical analyses were performed using Stata (version 16).

Multiple imputation by chained equations was used to produce 20 imputed data sets to minimize bias caused by missing data. Missing values were assumed to be random. Multiple imputation provides unbiased estimates in the presence of missing data under the missing at random assumption [25]. Details on the implementation of multiple imputation are provided in the S1.2 appendix in S1 Appendix. A complete case analysis was also conducted.

Finally, as additional sensitivity analysis, to limit noises at hospital level due to small sample size, we excluded from the sample hospitals with less than or up to 10 patients (37 hospitals corresponding to 179 observations) and replicated the PWE full model.

## Results

### Sample selection

In total, 14 742 patients with chronic HF were included in the HF-LT Registry from 247 hospitals, located in 37 countries, from 2011 to 2018, with most cases collected between 2011 and 2014. From this sample, we selected patients applying three criteria. First, we selected only patients whose country was included in the ATLAS registry and whose participating hospital submitted the ESC-HF-LT Site Questionnaire. Of the 37 nations included in the patient's registry, 31 are included in ATLAS and 188 hospitals had site information available. Combining these restrictions together, in order to have patients with information on both hospital and country, we derived 11 347 observations collected from 165 hospitals across 27 countries. Second, only patients whose vital status at 12 months was known were included, thus dropping 2027 cases with a sample of 9320 patients in 147 hospitals and 26 countries. Third, we discharged observations from countries in which less than 30 patients were observed (thus dropping 43 patients, in 5 hospitals, in 4 countries). Thus, the analytical cohort corresponded to 9277 patients from 142 hospitals in 22 countries. Fig 1 illustrates the sample selection process through a STROBE flow chart and S1 Table in the S1 Appendix shows sample reduction by country.

### Baseline characteristics

Tables 1 and 2 illustrate, respectively, the patient and hospital characteristics. S2 Table in the S1 Appendix illustrates descriptive statistics of patients for all available cases, before sample selection (N = 14 742), to show descriptively whether the selected sample presents remarkable differences compared to all patients enrolled. Selected cases presented very similar characteristics compared to the original cohort of patients included in the registry. Country's characteristics are reported in S1.3 Table of the S1 Appendix.

### Follow-up

Overall, the mean follow-up time was 342 days (sd 59.35, median 365, interquartile range (IQR) 356–365). 620 deaths were registered, with an all-cause mortality rate per 100 person-

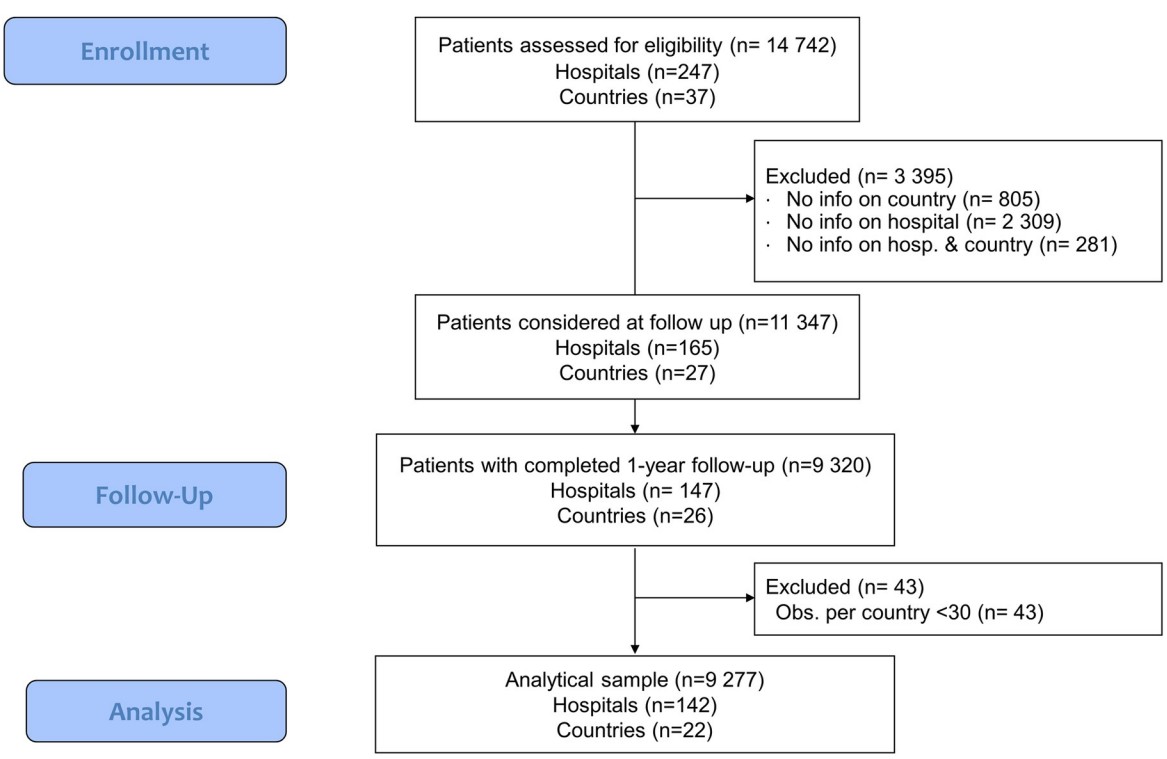

**Fig 1. STROBE flow chart.**

year equal to 7.1 (95% confidence interval (CI) 6.6–7.7). Fig 2 presents crude mortality rates and adjusted mortality rates with 95% CI for each country (we illustrate 95% CI of crude mortality rates in S1 Fig in S1 Appendix to avoid overcrowd the graph). When we look at the crude rates (green markers), remarkable differences in mortality rates across countries was observed, although confidence intervals were wide for some countries (see S1 Fig in S1 Appendix). Adjusted mortality rates were obtained from a Poisson model (using multiple imputation) adjusting for patient's age and sex and country (red markers) and all clinical and hospital variables (blue markers). The adjusted rates were often not statistically different, with wide CIs for some countries.

Fig 3 shows descriptively the variation in mortality rates observed across hospitals and countries. The units of analysis here are respectively hospitals (including only hospitals with at least 3 patients, to reduce outliers) (box plot on the left) and countries (box plot on the right). We observed a slightly larger variation in mortality rates between hospitals (median rate per 100-person year 7.8, IQR 5.2–12.4) compared to country (median 6.8, IQR 5.6–11.2). The same statistics were produced for adjusted mortality rates (not shown), and results were relatively similar for countries (median 7.0, IQR 5.7–8.8) and more variation was observed for hospitals (median 7.7, IQR 3.0–13.8).

## Mortality risk factors

Table 3 shows results from the PWE survival models (corresponding results based on complete case analysis are available in S4 Table of the S1 Appendix). Results are reported as incidence rate ratios. Estimates of incidence ratios (fixed effects) are consistent across multiple imputation and complete case analysis. Patient's characteristics are associated with 1-year mortality as expected: patients with older age (IRR = 1.03, 95% CI 1.02–1.04), suffering from diabetes

**Table 1. Baseline characteristics of the study population.**

| Variable | Observed | Missing values N (%) |
|---|---|---|
| Women | 2613 (28.2) | 1 (0.01) |
| Age in years. Mean (sd) | 65 (13.2) | 105 (1.1) |
| BMI, Kg/m$^2$. Mean (sd) | 28.1 (5.2) | 125 (1.3) |
| BMI <20 | 62 (0.7) | |
| BMI 20–24.9 | 2530 (27.3) | |
| BMI 25–29.9 | 3731 (40.2) | |
| BMI 30–34.9 | 1992 (21.5) | |
| BMI>35 | 837 (9.0) | |
| Systolic blood pressure, mmHg. Mean (sd) | 123.7 (20.9) | 29 (0.3) |
| Ischemic aetiology (%) | 3993 (43) | 6 (0.06) |
| Atrial fibrillation (%) | 3465 (37.4) | 9 (0.1) |
| Diabetes mellitus (%) | 2968 (32) | 5 (0.05) |
| Peripheral artery disease (%) | 1158 (12.5) | 31 (0.3) |
| Chronic obstructive pulmonary disease (%) | 1319 (14.2) | 18 (0.2) |
| Chronic kidney dysfunction (%) | 1790 (19.3) | 10 (0.1) |
| Implantable cardioverter defibrillator therapy (%) | 2621 (28.3) | 25 (0.3) |
| Left ventricular ejection fraction (%) | | |
| <40 | 5304 (57.2) | 599 (6.5) |
| 40–49 | 1663 (17.9) | |
| > = 50 | 1711 (18.4) | |
| New York Heart Association (NYHA) class III/IV (%) | 2377 (25.6) | 11 (0.1) |
| Pulmonary rales/hepatomegaly/peripheral oedema (%) | 2866 (30.9) | 18 (0.2) |
| Third heart sound (%) | 527 (5.7) | 39 (0.4) |
| Moderate or severe aortic stenosis (%) | 294 (3.2) | 1 336 (14.4) |
| Angiotensin-converting enzyme (ACE)-inhibitors (%) | 6189 (66.7) | 11 (0.1) |
| Angiotensin receptor blocker (ARB) (%) | 2350 (25.3) | 3 (0.03) |
| ACE inhibitors and/or ARBs (%) | 8225 (88.7) | 11 (0.1) |
| Beta Blockers (%) | 8273 (89.2) | 3 (0.03) |

% are based on total number of patients (n = 9 277).

Values are number (%) of total sample (N = 9277) unless stated otherwise.

mellitus (1.37, 1.15–1.63), PAD (1.56, 1.27–1.92), renal dysfunction (1.82, 1.52–2.18), with NYHA class III/IV (compared to I/II) (1.92, 1.60–2.30), having pulmonary or peripheral

**Table 2. Hospitals' baseline characteristics.**

| Variable | Observed | Missing N (%) |
|---|---|---|
| Catheterization laboratories (%) | 101 (71.1) | 18 (12.7) |
| Electrophysiological laboratories (%) | 82 (57.8) | 20 (14.1) |
| Cardiology Echocardiography service 24 hours on site (vs Regular hours) (%) | 65 (47.8) | 9 (6.3) |
| Cardiology Angiography/PCI service 24 hours on site (vs Regular hours) (%) | 51 (35.9) | 36 (25.4) |
| HF unit/clinic available for follow-up (%) | 89 (62.7) | 15 (10.6) |
| Heart transplant (%) | 55 (38.7) | 53 (37.3) |

% are based on total number of hospitals (n = 142).

Values are number (%) of total sample (N = 142) unless stated otherwise.

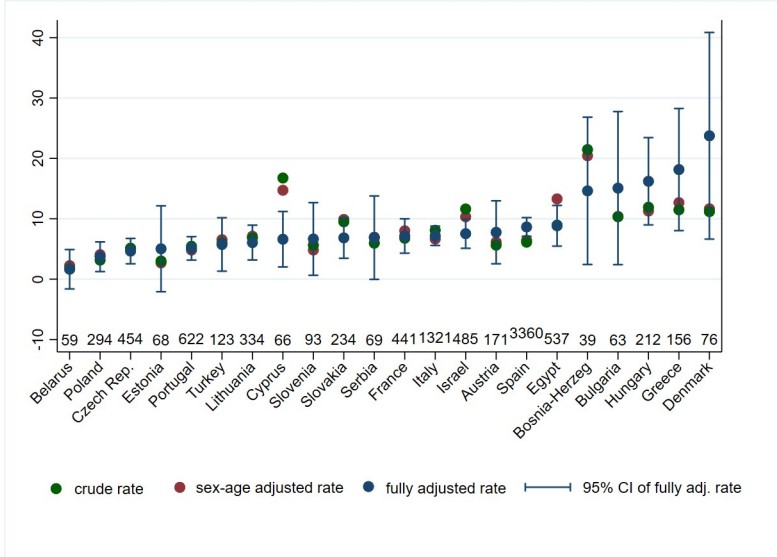

**Fig 2. Crude mortality rates and adjusted mortality rates with 95% CI (vertical blue lines), by country sorted by lowest to highest adjusted rate.**

congestion (1.85, 1.54–2.23) have higher mortality rate at one year; other patient's characteristics including left ventricular ejection fraction higher than 50 (0.78, 0.60–1.01), the use of ACE inhibitors or ARBs (0.71, 0.57–0.87) appeared to be associated with lower mortality rates at 1 year. Higher BMI (e.g. IRR for BMI between 25 and 30 vs BMI<20 equal to 0.45, 0.21–0.95) and higher systolic blood pressure (0.92, 0.90–0.95) also resulted associated with lower mortality, in line with previous findings [11]. In terms of hospital characteristics, only dedicated HF clinic was significantly associated with a lower mortality rate, by around 40% (0.64, 0.46–0.89). No other hospital characteristics were found to be related to 1-year mortality, neither the selected country characteristics.

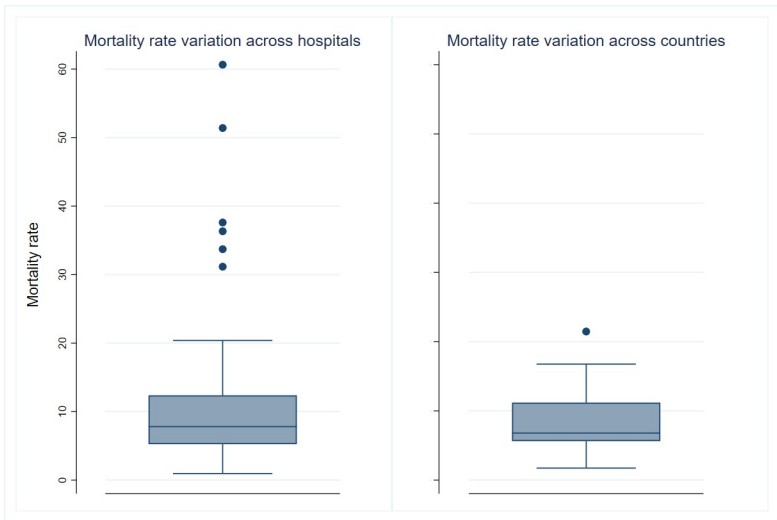

**Fig 3. Boxplot of the crude mortality rates per 100-person year across hospitals and countries.**

**Table 3. Factor associated with 1-year all-cause mortality for patients with chronic HF, PWE survival model, incidence rate ratios and 95% CI.**

| | Patient's characteristics | Patient + Hospital characteristics | Patient + Country characteristics | Patient + Hospital + Country characteristics |
|---|---|---|---|---|
| | IRR (95% CI) | IRR (95% CI) | IRR (95% CI) | IRR (95% CI) |
| *Clinical variables* | | | | |
| Males (vs females) | 1.23** (1.01–1.51) | 1.24** (1.02–1.52) | 1.23** (1.01–1.51) | 1.24** (1.01–1.52) |
| Age in years | 1.03*** (1.02–1.04) | 1.0*** (1.02–1.04) | 1.0*** (1.02–1.04) | 1.03*** (1.02–1.04) |
| BMI (Kg/m$^2$) (20–24.9) vs BMI<20 | 0.70 (0.335–1.47) | 0.69 (0.33–1.45) | 0.71 (0.34–1.48) | 0.71 (0.34–1.48) |
| BMI (Kg/m$^2$) (25–29.9) vs BMI<20 | 0.45** (0.21–0.94) | 0.44** (0.21–0.92) | 0.45** (0.21–0.95) | 0.45** (0.21–0.95) |
| BMI (Kg/m$^2$) (30–34.9) vs BMI<20 | 0.43** (0.20–0.92) | 0.42** (0.20–0.89) | 0.44** (0.20–0.93) | 0.43** (0.20–0.93) |
| BMI (Kg/m$^2$) (> = 35) vs BMI<20 | 0.43** (0.19–0.95) | 0.42** (0.19–0.94) | 0.44** (0.20–0.97) | 0.44** (0.20–0.98) |
| Systolic blood pressure, mmHg | 0.93*** (0.91–0.95) | 0.92*** (0.90–0.95) | 0.93*** (0.91–0.95) | 0.92*** (0.90–0.95) |
| Ischemic etiology | 1.10 (0.92–1.31) | 1.11 (0.93–1.32) | 1.10 (0.92–1.31) | 1.11 (0.93–1.33) |
| Atrial Fibrillation history | 1.10 (0.93–1.30) | 1.11 (0.93–1.31) | 1.10 (0.93–1.31) | 1.11 (0.94–1.31) |
| Diabetes history | 1.35*** (1.14–1.61) | 1.36*** (1.14–1.62) | 1.36*** (1.15–1.62) | 1.37*** (1.15–1.63) |
| Peripheral artery disease | 1.54*** (1.25–1.89) | 1.56*** (1.27–1.91) | 1.55*** (1.26–1.91) | 1.56*** (1.27–1.92) |
| Chronic obstructive pulmonary disease | 1.20* (0.98–1.47) | 1.20* (0.98–1.46) | 1.20* (0.98–1.47) | 1.19* (0.98–1.46) |
| Chronic kidney dysfunction | 1.80*** (1.50–2.15) | 1.80*** (1.50–2.15) | 1.81*** (1.52–2.17) | 1.82*** (1.52–2.18) |
| Implantable cardioverter defibrillator therapy | 0.96 (0.79–1.17) | 0.97 (0.80–1.18) | 0.99 (0.81–1.21) | 0.99 (0.81–1.21) |
| New York Heart Association (NYHA) (III/IV vs I/II) | 1.90*** (1.59–2.28) | 1.90*** (1.59–2.28) | 1.91*** (1.59–2.29) | 1.92*** (1.60–2.30) |
| Peripheral oedema/pulmonary rale | 1.87*** (1.56–2.24) | 1.84*** (1.53–2.21) | 1.84*** (1.53–2.22) | 1.85*** (1.54–2.23) |
| s3gallop | 1.22 (0.91–1.65) | 1.19 (0.88–1.60) | 1.17 (0.87–1.58) | 1.17 (0.86–1.58) |
| Medication Angiotensin-converting enzyme (ACE)/ Angiotensin receptor blocker (ARB) | 0.71*** (0.58–0.88) | 0.71*** (0.57–0.88) | 0.70*** (0.57–0.87) | 0.71*** (0.57–0.87) |
| Medication beta blocker | 0.82* (0.64–1.03) | 0.83 (0.66–1.06) | 0.84 (0.66–1.07) | 0.85 (0.67–1.08) |
| Left ventricular ejection fraction (EF) 40–49 (vs EF<40) | 0.83 (0.65–1.06) | 0.83 (0.65–1.06) | 0.83 (0.65–1.06) | 0.83 (0.65–1.06) |
| EF> = 50 (vs EF<40) | 0.76** (0.59–0.98) | 0.77** (0.60–0.99) | 0.77** (0.60–0.99) | 0.78* (0.60–1.01) |
| Interval 90–180 days | 1.16 (0.93–1.46) | 1.16 (0.93–1.46) | 1.16 (0.93–1.46) | 1.17 (0.93–1.46) |
| 180–270 days | 1.26** (1.01–1.58) | 1.26** (1.01–1.58) | 1.26** (1.01–1.58) | 1.26** (1.007–1.58) |
| 270–365 days | 1.30** (1.03–1.63) | 1.30** (1.03–1.63) | 1.29** (1.03–1.63) | 1.30** (1.03–1.63) |
| *Hospital's characteristics* | | | | |
| Catheterisation lab | | 1.30 (0.86–1.98) | | 1.37 (0.88–2.13) |
| Electrophysiological lab | | 0.96 (0.66–1.40) | | 0.94 (0.64–1.40) |
| Cardiology Echocardiography service 24 hrs on site (vs Regular hours) | | 1.01 (0.76–1.34) | | 0.98 (0.72–1.33) |
| Cardiology Angiography/PCI service 24 hrs on site (vs Regular hours) | | 0.93 (0.69–1.26) | | 0.95 (0.70–1.31) |
| Heart failure unit | | 0.62*** (0.46–0.83) | | 0.64*** (0.46–0.89) |
| Heart transplantation | | 0.87 (0.64–1.18) | | 0.84 (0.61–1.16) |
| *Country's characteristics* | | | | |
| Gross Domestic Product (GDP) (in thousands $) | | | 0.99 (0.93–1.05) | 1.00 (0.94–1.07) |
| Life Expectancy at birth (years) | | | 0.93 (0.77–1.13) | 0.94 (0.77–1.14) |
| Total health expenditure (% of GDP) | | | 1.10 (0.94–1.27) | 1.11 (0.94–1.30) |
| Gini Index | | | 3.33 (0.001–12,146.57) | 17.83 (0.01–59,194.69) |
| Health expenditure per capita (PPP) (in thousands $) | | | 1.112 (0.719–1.722) | 1.12 (0.72–1.74) |
| Cardiovascular disease deaths per million inhabitants (in thousands per year) | | | 0.997 (0.803–1.238) | 1.053 (0.849–1.307) |
| Health system Bismarck (vs Beveridge) | | | 1.213 (0.548–2.682) | 1.138 (0.515–2.516) |
| Health system Semashko (vs Beveridge) | | | 1.090 (0.424–2.804) | 0.883 (0.349–2.232) |
| **Random Effects** | | | | |

*(Continued)*

**Table 3.** (Continued)

| | Patient's characteristics | Patient + Hospital characteristics | Patient + Country characteristics | Patient + Hospital + Country characteristics |
|---|---|---|---|---|
| | IRR (95% CI) | IRR (95% CI) | IRR (95% CI) | IRR (95% CI) |
| Country variance (s.e.)§ (95% CI) | 2.28e-06 (.03769) (0.000 -.) | 5.42e07 (.3467) (0.000 -.) | 4.97e-07 (.0752) (0.000 -.) | 1.83e-06 (.5788) (0.000 -.) |
| Hospital variance (95% CI) | 0.456 (0.322–0.646) | 0.393 (0.263–0.586) | 0.406 (0.278–0.594) | 0.372 (0.247–0.561) |
| Observations | 35,812 | 35,812 | 35,812 | 35,812 |
| Number of hospitals | 142 | 142 | 142 | 142 |
| Number of countries | 22 | 22 | 22 | 22 |

IRR = incidence rate ratio; CI = confidence interval

*** p<0.01

** p<0.05

* p<0.1.

§ s.e. reported given 95% CI was not computable.

Looking at the variance component (random effects, bottom of Table 3), variation in mortality due to differences between countries was negligible across all model specifications (1.83e-06 in the model adjusted for all selected variables), and variation between hospitals was evident in each model specification, and it reduced by about 20% following adjustment for hospital-level and country-level variables (0.456 when accounting only for patient's characteristics, 0.372 adjusting for all-level variables).

Comparing these results with those obtained from complete case analysis (S4 Table in S1 Appendix), we observed that between-hospital variance was smaller in complete case analysis (0.030, 95% CI 4.95E-06–12.622 in the model adjusted for all-level variables) compared to results obtained using multiple imputation, where more observations (patients, hospitals, countries) were included in the model. This may suggest a selection of more homogeneous hospitals in the complete case analysis and a consequent reduction of variation between centres. Between-country variance was unstable across model specifications in complete case analysis and generally lower than between-hospital variance.

Results were replicated within a discrete time survival model (see S1 Appendix), for both multiple imputation and complete case analysis (S5 Table in S1 Appendix). Estimates are reported as odds ratios and hence not directly comparable, but significance, direction and magnitude of results are consistent across model specifications, comparing multiple imputation and complete case analyses respectively. From the multilevel logistic regression model, we found that after case mix adjustment the mortality risk of patients in the same country was basically uncorrelated (ICC 9.156e-07), whilst mortality risk for patients within the same hospital had a higher correlation (ICC = 0.084).

## Discussion

### Principal findings

In this international cohort of 9277 participants from 22 countries which pooled retrospective data from a bespoke patient and hospital datasets and ATLAS of cardiology, there was evidence that clinical variables, shown by different studies to be associated with mortality of patients with chronic HF, have a significant role in the association with HF outcomes also after adjustment for hospital and country variables.

The 1-year all-cause mortality rate per 100 person-year was equal to 7.1 (95% CI 6.6–7.7), largely varying across countries and hospitals. We identified factors related to hospital and country variations and accounting for these variables, we found almost null between-country variance and higher between-hospital variance. We explored the additive value of hospital and country characteristics to clinical characteristics and it emerged that the presence of dedicated HF clinics was the strongest predictor of all country and hospital specific variables. This represents a key contribution of this work, as discussed below.

## Comparison with other studies

Our study supports the set of prognostic clinical variables already emerged as statistically significant in other studies, [26–30] but with the uniqueness to have included in the adjusted model hospital and country variables. This observation reinforces the relevance of the clinical characteristics of the patients to be considered for prognostic purposes, irrespective of both borders and typology of hospitals.

Among hospital characteristics, the availability of a dedicated HF clinic appeared to be independently associated with a lower risk of all-cause death. The complexity of the management of patients with HF, generally old, with multiple comorbidities, with the need to assume daily several drugs needs the activity and the expertise of several health professional. The ideal setting for establishing all the diagnostic and therapeutic strategies recommended for these complex patients seems to be the HF clinic where the possibility to have an easy access to nurses and physicians with expertise in HF and in the treatment of the most frequent comorbidities can be assured [31]. The multi-professional model of care provided by a dedicated HF clinic was demonstrated to be able to improve adherence to treatments recommended by current guidelines and to significantly reduce the need of hospital admissions and mortality [32–35]. Our study confirms the relevant role of these HF dedicated hospital structures in a multinational setting and even after the adjustment for the clinical, hospital and country characteristics.

To interpret the lack of association between country level factors and mortality among patients with chronic HF as well as the low country variation, it is useful to put the results in a global perspective. Among European high-income countries, that compose the majority of our sample, the socio-economical differences may not be strong enough to impact measurably on survival. By contrast in the low and low-middle income countries, it has been found that two third of attributable risk is carried by socio-economic factors [36,37]. A study focused on low-middle income countries found marked regional differences in mortality in patients with HF that persisted even after multivariable adjustment for cardiac and non-cardiac factors [8]; while a research that considered both middle and high income countries, found that after accounting for clinical characteristics of patients only Latin America and Pacific Asia presented higher mortality rates compared to the other sub-continents [9].

The results of our work shed further light in the literature on geographic variation in HF outcomes, because it focuses on European and non-European Mediterranean countries considered individually (rather than as subcontinental clusters) and especially because it considers also risk factors measured at hospital and country level, not only clinical variables.

## Strengths and limitations of study

The main strength of this research comes from the use of multisource linked data that allowed us to explore the concurrent role of clinical conditions of participants and hospital's and country's characteristics on the outcomes of patients with chronic HF. The determinants of outcomes of patients with HF have been evaluated by several groups but, in most cases, only

considering clinical variables. This study built on existing literature and expanded current knowledge by taking into consideration other relevant aspects that could affect the clinical course of this population of patients, the hospital characteristics and some variables describing the health care and socioeconomic context in which patients live and received treatments.

The fact that the study design had a mandatory follow-up visit at 12 months to collect information on morbidity and mortality represented a strength and unique opportunity to study survival with a relatively uniform follow-up time across countries. At the same time, however, we had no information on the time course of patients' HF over this period. Not being able to monitor the HF progress, little we know on possible precipitating factors that may lead to decompensation during the follow-up, and that are known to be different in young and older patients [38]. Nevertheless, we accounted for patients' clinical and medical history, partly accounting for those factors that may act as precipitators in the HF time course.

This study suffers from some specific limitations that should be mentioned. The first one concerns sample selection and generalizability of our findings. The National Cardiac Societies which participated in the ESC registry were requested to appoint a national coordinator responsible for creating a network of centres and to manage it in connection with the EORP team. However, while a central data quality control was systematic and accurate, a systematic peripheral auditing was not possible, thereby the compliance to the protocol recommendations was not individually verified. This implies a threefold limitation: not all countries contributed to the questionnaire equally in terms of i) number of patients submitted, ii) type of patients i.e. case severity mix and iii) types of participating hospitals. Moreover, the fact that the selection of hospitals was made by ESC guarantees a recognized quality of participating hospitals, but at the same time entails a possible selection bias. This altered how representative a country's contribution was to the registry and therefore when we referred to each country, we in fact referred to its contribution to the registry. This contribution varied between countries leading to uneven representation of countries.

Several observations were discharged because no information on the participating hospital characteristics was available affecting the generalizability of results. Within the sample selected, item nonresponse was addressed using multiple imputation. Compared to complete case analysis, fixed effects results (coefficients estimate) were similar, while the random part, capturing the country and hospital variation, revealed higher between-hospital variability when using all selected observations. This warns on the risk of selection bias, as highlighted among the study limitations, because responding centres might represent a subsample of homogeneous hospitals, hence not representative of all enrolled centres. Such limitation must be carefully considered to interpret the key finding of this study on the protective effect of the presence of a HF clinic: for 30% of the centres that took part to the registry, information on HF clinics (n = 76 out of 247) was missing; among those selected for this study 10% did not report this information (15 out of 142), but we were able to impute it and the evidence of a strong association with mortality was found both using multiple imputation and complete case analysis. Further research is needed to confirm the generalizability of our findings. Finally, we also performed a sensitivity analysis dropping observations from hospitals with less than or up to 10 patients to reduce noises due to small sample size of clusters and obtained results (S6 Table in S1 Appendix) very similar to those presented in Table 3 (column 4).

## Conclusions and policy implications

A unique contribution of this study is that it considered risk factors for HF outcomes acting at three levels simultaneously: clinical, hospital and country. Results confirmed that the clinical variables known to be associated with outcomes of patients with HF remained significantly

associated with mortality even after adjustment for hospital and country characteristics. The finding that the presence, at hospital level, of dedicated HF clinics was significantly associated with a lower mortality rate suggests that this type of health care service can probably assure a more appropriate treatment of patients and a higher adherence to the recommendations of the current ESC guidelines, which recommend that patients with HF are enrolled in a multidisciplinary care management programmes. [3] Such an approach has been found reducing HF hospitalization and mortality in patients discharged from the hospital. [32,33] Our work contributes to this discussion providing empirical evidence on the protective effect of HF clinics to support this statement and may be used to strengthen this statement in future guidelines.

The ESC guidelines suggest that "content and structure of HF management programmes may vary in different countries and health care settings". Due to the data limitations we cannot provide further insight on country variability in this matter, and further research is needed to intersect the role of hospital facilities with country's health care settings.

The other contribution of this research pertains to the assessment of country and hospital variation in mortality. The finding that after the full adjustment for clinical, hospital and country variables, there appeared to be some variation only at hospital level suggests that, in a cardiology setting of ESC member countries, presumably there remain a certain level of variability in the adoption of guideline recommendations, contributing to inequalities in patients' outcomes. The timely and full adherence to clinical practice guidelines is fundamental for assuring high level of care. Efforts should be placed to guarantee their applications across and within ESC member countries and enforce a close monitoring.

## Supporting information

**S1 Appendix.**
(DOCX)

**S1 File. Patient consent form.**
(PDF)

**S2 File. Case report form of the Long-Term Registry on patients with heart failure.**
(PDF)

## Author Contributions

**Conceptualization:** Benedetta Pongiglione, Aleksandra Torbica, Aldo P. Maggioni.

**Data curation:** Benedetta Pongiglione.

**Formal analysis:** Benedetta Pongiglione.

**Investigation:** Benedetta Pongiglione, Aleksandra Torbica, Aldo P. Maggioni.

**Methodology:** Benedetta Pongiglione.

**Project administration:** Aldo P. Maggioni.

**Supervision:** Aleksandra Torbica, Chris P. Gale, Luigi Tavazzi, Panos Vardas, Aldo P. Maggioni.

**Visualization:** Benedetta Pongiglione, Chris P. Gale, Luigi Tavazzi, Panos Vardas, Aldo P. Maggioni.

**Writing – original draft:** Benedetta Pongiglione, Aleksandra Torbica, Chris P. Gale, Luigi Tavazzi, Aldo P. Maggioni.

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
