## [Decision Letter · Decision Letter 0]

9 Mar 2021

PONE-D-21-03871

Patient, hospital and country-level risk factors of all-cause mortality among patients with chronic heart failure: Prospective international cohort study

PLOS ONE

Dear Dr. PONGIGLIONE,

Thank you for submitting your manuscript to PLOS ONE. After careful consideration, we feel that it has merit but does not fully meet PLOS ONE’s publication criteria as it currently stands. Therefore, we invite you to submit a revised version of the manuscript that addresses the points raised during the review process.

We look forward to receiving your revised manuscript.

Kind regards,

Pasquale Abete

Academic Editor

PLOS ONE

Additional Editor Comments:

The manuscript is very interesting and topic. However I have only a concern about the importance of HF-related precipitating factors. Please see and discuss Testa G et al. Precipitating factors in younger and older adults with decompensated chronic heart failure: are they different? J Am Geriatr Soc. 2013 Oct;61(10):1827-8.

Journal Requirements:

2. Thank you for including your ethics statement:  "This study complies with the Declaration of Helsinki. Participation in the ESC-HF-LT-R had been approved by each local institutional review board in accordance with its country’s legislation. All participants provided written informed consent. No data were collected before the patient received detailed information and gave signed informed consent.".   

Please amend your current ethics statement to include the full name of the ethics committee that approved your specific study.

For additional information about PLOS ONE submissions requirements for ethics oversight of animal work, please refer to http://journals.plos.org/plosone/s/submission-guidelines#loc-animal-research  

Reviewers' comments:

Reviewer's Responses to Questions

**Comments to the Author**

1. Is the manuscript technically sound, and do the data support the conclusions?

Reviewer #1: Yes

Reviewer #2: Yes

2. Has the statistical analysis been performed appropriately and rigorously? 

Reviewer #1: Yes

Reviewer #2: Yes

3. Have the authors made all data underlying the findings in their manuscript fully available?

Reviewer #1: Yes

Reviewer #2: No

4. Is the manuscript presented in an intelligible fashion and written in standard English?

Reviewer #1: Yes

Reviewer #2: Yes

5. Review Comments to the Author

Reviewer #1: In the manuscript entitled Patient, hospital and country-level risk factors of all-cause mortality among patients with chronic heart failure: Prospective international cohort study, Pongiglione and co-authors report on patient-, hospital- and country-level characteristics associated with 1-year all-cause mortality among patients with chronic HF including geographic and hospital variation in mortality. On 9,277 patients with chronic HF in the prospective cohort study European Society of Cardiology Heart Failure Long Term, the Authors report an all-cause 1-year mortality rate, across 142 hospitals located in 22 countries, of 7.1% which varied between countries and hospitals. These mortality rates were directly associated with increasing age, diabetes, peripheral artery disease, higher NYHA class. Conversely, treatment with angiotensin-converting enzyme inhibitor and angiotensin receptor antagonists and being managed in an HF clinic were found protective against mortality. Interestingly, no other hospital-level characteristics, and no country-level healthcare characteristics were associated with 1-year mortality, suggesting that between-hospital variance might be crucial for HF patients’ outcomes.

Overall, this is a very interesting manuscript on a crucial topic in HF management. The aim is clear, data analysis is well conducted and reported, and results are well discussed. I consider the manuscript acceptable for publication as it stands.

Reviewer #2: The Authors explored hospital and country-level characteristics associated with 1-year all-cause mortality among patients with chronic HF and investigates geographic and hospital variation in mortality. They studied 9,277 patients with chronic HF enrolled between May 2011 and November 2017 in the prospective cohort study European Society of Cardiology Heart Failure Long Term registry across 142 hospitals, located in 22 countries. The mean age of the selected outpatients was 65 years and the all-cause 1- year mortality rate per 100 person-years was 7.1 and varied between countries (median 6.8, IQR 5.6-11.2) and hospitals (median 7.8, IQR 5.2-12.4). Mortality was associated with age (incidence rate ratio 1.03, 95% CI 1.02-1.04), diabetes mellitus (1.37, 1.15-1.63), peripheral artery disease (1.56, 1.27- 1.92), New York Heart Association class III/IV (1.91, 1.60-2.30), treatment with angiotensin-converting enzyme inhibitor and angiotensin receptor antagonists (0.71, 0.57-0.87) and HF clinic (0.64, 0.46-0.89). No other hospital-level characteristics, and no country-level healthcare characteristics were associated with 1-year mortality, with case-mix standardised variance between countries being very low and higher for hospitals (0.372).

I find the study of great interest, data support conclusion even if I suppose a possible role of a selection bias due to the selection made by European Society of Cardiology in the selection of hospital. The involved hospital are of recognize high quality and specifically devoted to cardiovascular disease. This reflect the great Job made the ESC in the diffusion of cardiovascular knowledge all over the Europe.

6. PLOS authors have the option to publish the peer review history of their article (what does this mean?). If published, this will include your full peer review and any attached files.

Reviewer #1: No

Reviewer #2: **Yes: **Francesco CACCIATORE

---

## [Author Response · Author response to Decision Letter 0]

29 Mar 2021

Additional Editor Comments:

The manuscript is very interesting and topic. However I have only a concern about the importance of HF-related precipitating factors. Please see and discuss Testa G et al. Precipitating factors in younger and older adults with decompensated chronic heart failure: are they different? J Am Geriatr Soc. 2013 Oct;61(10):1827-8.

Authors: We thank the Associate Editor for this comment. We discussed the importance of HF-related precipitating factors in the discussion and discussed the suggested paper by Testa et al. The paragraph that we added is the following 

“The fact that the study design had a mandatory follow-up visit at 12 months to collect information on morbidity and mortality represented a strength and unique opportunity to study survival with a relatively uniform follow-up time across countries. At the same time, however, we had no information on the time course of patients’ HF over this period. Not being able to monitor the HF progress, little we know on possible precipitating factors that may lead to decompensation during the follow-up, and are known to be different in young and older patients (Testa et al., 2013). Nevertheless, we accounted for patients’ clinical and medical history, partly accounting for those factors that may act as precipitators in the HF time course.”

Review Comments to the Author:

Reviewer #1: In the manuscript entitled Patient, hospital and country-level risk factors of all-cause mortality among patients with chronic heart failure: Prospective international cohort study, Pongiglione and co-authors report on patient-, hospital- and country-level characteristics associated with 1-year all-cause mortality among patients with chronic HF including geographic and hospital variation in mortality. On 9,277 patients with chronic HF in the prospective cohort study European Society of Cardiology Heart Failure Long Term, the Authors report an all-cause 1-year mortality rate, across 142 hospitals located in 22 countries, of 7.1% which varied between countries and hospitals. These mortality rates were directly associated with increasing age, diabetes, peripheral artery disease, higher NYHA class. Conversely, treatment with angiotensin-converting enzyme inhibitor and angiotensin receptor antagonists and being managed in an HF clinic were found protective against mortality. Interestingly, no other hospital-level characteristics, and no country-level healthcare characteristics were associated with 1-year mortality, suggesting that between-hospital variance might be crucial for HF patients’ outcomes.

Overall, this is a very interesting manuscript on a crucial topic in HF management. The aim is clear, data analysis is well conducted and reported, and results are well discussed. I consider the manuscript acceptable for publication as it stands.

Authors: We thank the reviewer very much for her/his positive and supportive feedback.

Reviewer #2: The Authors explored hospital and country-level characteristics associated with 1-year all-cause mortality among patients with chronic HF and investigates geographic and hospital variation in mortality. They studied 9,277 patients with chronic HF enrolled between May 2011 and November 2017 in the prospective cohort study European Society of Cardiology Heart Failure Long Term registry across 142 hospitals, located in 22 countries. The mean age of the selected outpatients was 65 years and the all-cause 1- year mortality rate per 100 person-years was 7.1 and varied between countries (median 6.8, IQR 5.6-11.2) and hospitals (median 7.8, IQR 5.2-12.4). Mortality was associated with age (incidence rate ratio 1.03, 95% CI 1.02-1.04), diabetes mellitus (1.37, 1.15-1.63), peripheral artery disease (1.56, 1.27- 1.92), New York Heart Association class III/IV (1.91, 1.60-2.30), treatment with angiotensin-converting enzyme inhibitor and angiotensin receptor antagonists (0.71, 0.57-0.87) and HF clinic (0.64, 0.46-0.89). No other hospital-level characteristics, and no country-level healthcare characteristics were associated with 1-year mortality, with case-mix standardised variance between countries being very low and higher for hospitals (0.372).

I find the study of great interest, data support conclusion even if I suppose a possible role of a selection bias due to the selection made by European Society of Cardiology in the selection of hospital. The involved hospital are of recognize high quality and specifically devoted to cardiovascular disease. This reflect the great Job made the ESC in the diffusion of cardiovascular knowledge all over the Europe.

Authors: We thank the reviewer very much for her/his positive feedback. We included the comment on the ESC role in selecting hospitals in the discussion in the following paragraph:

(..) Moreover, the fact that the selection of hospitals was made by ESC guarantees a recognized quality of participating hospitals, but at the same time entails a possible selection bias.

References:

Authors: We added the reference 

Testa G, Della‐Morte D, Cacciatore F, Gargiulo G, D'Ambrosio D, Galizia G, et al. Precipitating factors in younger and older adults with decompensated chronic heart failure: are they different?. Journal of the American Geriatrics Society. 2013 Oct;61(10):1827-8.

And amended those not compliant with Vancouver form as described in Plos submission guidelines

---

## [Decision Letter · Decision Letter 1]

19 Apr 2021

Patient, hospital and country-level risk factors of all-cause mortality among patients with chronic heart failure: Prospective international cohort study

PONE-D-21-03871R1

Dear Dr. PONGIGLIONE,

We’re pleased to inform you that your manuscript has been judged scientifically suitable for publication and will be formally accepted for publication once it meets all outstanding technical requirements.

Kind regards,

Pasquale Abete

Academic Editor

PLOS ONE

Additional Editor Comments (optional):

No further comments.

Reviewers' comments:

Reviewer's Responses to Questions

**Comments to the Author**

1. If the authors have adequately addressed your comments raised in a previous round of review and you feel that this manuscript is now acceptable for publication, you may indicate that here to bypass the “Comments to the Author” section, enter your conflict of interest statement in the “Confidential to Editor” section, and submit your "Accept" recommendation.

Reviewer #1: All comments have been addressed

Reviewer #2: All comments have been addressed

2. Is the manuscript technically sound, and do the data support the conclusions?

Reviewer #1: Yes

Reviewer #2: Yes

3. Has the statistical analysis been performed appropriately and rigorously? 

Reviewer #1: Yes

Reviewer #2: Yes

4. Have the authors made all data underlying the findings in their manuscript fully available?

Reviewer #1: Yes

Reviewer #2: Yes

5. Is the manuscript presented in an intelligible fashion and written in standard English?

Reviewer #1: Yes

Reviewer #2: Yes

6. Review Comments to the Author

Reviewer #1: The Authors addressed all the comments form the reviewers. I confirm that the manuscript is acceptable for publication as it stands.

Reviewer #2: (No Response)

7. PLOS authors have the option to publish the peer review history of their article (what does this mean?). If published, this will include your full peer review and any attached files.

Reviewer #1: No

Reviewer #2: No

---

## [Editor Report · Acceptance letter]

26 Apr 2021

PONE-D-21-03871R1 

Patient, hospital and country-level risk factors of all-cause mortality among patients with chronic heart failure: Prospective international cohort study 

Dear Dr. Pongiglione:

I'm pleased to inform you that your manuscript has been deemed suitable for publication in PLOS ONE. Congratulations! Your manuscript is now with our production department. 

Kind regards, 

on behalf of

Prof. Pasquale Abete 

Academic Editor

PLOS ONE